# Everyday Evaluation of Herb/Dietary Supplement–Drug Interaction: A Pilot Study

**DOI:** 10.3390/medicines10030020

**Published:** 2023-02-28

**Authors:** Joao Victor Souza-Peres, Kimberly Flores, Bethany Umloff, Michelle Heinan, Paul Herscu, Mary Beth Babos

**Affiliations:** 1DeBusk College of Osteopathic Medicine, Lincoln Memorial University, Harrogate, TN 37752, USA; 2School of Medical Sciences, Lincoln Memorial University, Harrogate, TN 37752, USA; 3Research Division, Herscu Laboratory, Amherst, MA 01002, USA

**Keywords:** herb–drug interaction, pharmacovigilance, pharmacoepidemiology, phytovigilance, drug interaction, herbal medicine

## Abstract

A lack of reliable information hinders the clinician evaluation of suspected herb–drug interactions. This pilot study was a survey-based study conceived as a descriptive analysis of real-life experiences with herb–drug interaction from the perspective of herbalists, licensed health-care providers, and lay persons. Reported dietary supplement–drug interactions were evaluated against the resources most commonly cited for the evaluation of potential supplement–drug interactions. Disproportionality analyses were performed using tools available to most clinicians using data from the U.S. Federal Adverse Event Reporting System (FAERS) and the US Center for Food Safety and Applied Nutrition (CFSAN) Adverse Event Reporting System (CAERS). Secondary aims of the study included exploration of the reasons for respondent use of dietary supplements and qualitative analysis of respondent’s perceptions of dietary supplement–drug interaction. While agreement among reported supplement–drug interactions with commonly cited resources for supplement–drug interaction evaluation and via disproportionality analyses through FAERS was low, agreement using data from CAERS was high.

## 1. Introduction

Self-care with herbal medicines is rapidly increasing around the world. The US Centers for Disease Control’s National Health and Nutrition Examination Survey (NHANES) found that almost 25% of respondents over the age of 60 years reported using four or more supplements daily [1].

As stated in the Dietary Supplement Health and Education Act (DSHEA) of 1994, the National Institutes of Health (NIH) defines a dietary supplement as a non-tobacco product intended to supplement the diet. Such a supplement contains one or more dietary ingredients (including vitamins, minerals, herbs or other botanicals, amino acids, and other substances), is intended to be taken by mouth, and is labeled on the front panel as being a dietary supplement [2]. This broad definition encompassed more than 80,000 products available for sale in the United States in 2018 [3].

Dietary supplements of botanical or herbal origin account for nearly 30% of all dietary supplements on the market, whether they contain extracts from a single plant or from multiple plant sources [3]. The botanical dietary supplement industry in the United States is expanding rapidly. In 2020, retail sales of herbal dietary supplements in the US exceeded $12 billion, representing a nearly 10% increase from 2019 [4]. Several thousand new botanical dietary supplement formulations enter the market each year [5].

In their analysis of the National Consumer Survey on the Medication Experience and Pharmacist Role (NCSME-PR) involving over 25,000 adults in the United States, Rashrash et al. discovered that nearly 35% of respondents reported currently using botanical dietary supplements [6]. Nearly 70% of those reporting use of herbal dietary supplements reported concomitant prescription medication use, and nearly 64% reported use of over-the-counter medications. Respondents with comorbid states including cancer, diabetes, heart disease, pulmonary disease, obesity, arthritis, and history of stroke reported a significantly higher prevalence of herbal dietary supplement use than those without illness, increasing the likelihood of dietary supplement in conjunction with prescription medications.

An increased prevalence of dietary supplement use by those with illness has been confirmed in several recent studies [7,8,9,10]. In their 2022 study, Prely et al. reported that 40–83% of oncology patients use complementary and alternative medicines during cancer treatment, particularly herbal dietary supplements [9]. In a survey of 806 oncology patients, Alsanad et al. reported that 53.7% were taking combinations of dietary supplements and prescription medications [11]. The authors identified 167 potential supplement–drug interactions that affected nearly 14% of respondents; the potential for herbal dietary supplement interaction with narrow therapeutic index cancer agents is particularly alarming [12].

According to the United States Food and Drug Administration (US FDA), tens of millions of United States adults take herbal dietary supplements concomitantly with prescription medications [13]. Concomitant herbal dietary supplement use by those taking prescription medication ranges from 20 to 35%, which poses a significant potential risk of clinically relevant interactions between medications and dietary supplements [14]. It is estimated that up to 70% of patients using dietary supplements do not report such use to their health-care provider, thereby decreasing the potential for prevention of harm from such interactions [15]. Furthermore, many health-care providers are unaware of the potential for interaction, which is due in part to a dearth of clinically relevant evidence [8,14,16,17], variability of data reported in herb–drug interaction databases [9,18,19], and lack of training in this area [14,16]. In their 2022 investigation of ten popular herbs’ interactions with cancer treatment drugs as reported to the World Health Organization’s adverse event reporting database known as Vigibase, Pochet et al. estimated that 5% of reported herbal dietary supplement–drug interactions could have been avoided if a reliable published source was available [20].

Evaluation of potential herb–drug interactions is fraught with difficulty. Under the DSHEA of 1994, the US FDA is not authorized to approve dietary supplements for safety or efficacy [21], although manufacturers must provide “reasonable safety data” for products introduced to market after 15 October 1994 [17]. It is important to note that such data are not required for products introduced prior to the adoption of the DSHEA [14]. The US FDA bears the burden of proof to deem a specific product is unsafe; however, since 2007, manufacturers of dietary supplements are required to report serious adverse effects to the US FDA, and manufacturers of dietary supplements are required to follow current good manufacturing practices (cGMP) guidelines [22]. The cGMP guidelines aim to ensure overall product quality by setting standards for accuracy of labeling, minimal standards for manufacturing, guaranteed absence of certain (but not all) contaminants, and availability of records for inspection. However, the guidelines are nonbinding on the manufacturer, and botanical supplements may vary substantially in composition [23]. A DNA-based analysis of 44 herbal products from 12 companies performed by Newmaster et al. found that 59% of the samples contained DNA from plant species not listed on the label, nearly 33% contained contaminants or fillers not listed on the label, and product substitution was found in over 65% of tested samples [24]. In a recent review of 107 case reports of herb–drug interactions, contaminant causality was likely in at least one case [25]. Additionally, the US DSHEA prohibits dietary supplement labeling from claims related to disease treatment or diagnosis, products are often labeled with vague monikers such as “thyroid support formula”. This further complicates the interpretation of case reports of HDI.

Notwithstanding the variability introduced through adulteration and misbranding, the nature of botanical sources themselves contribute to the substantial variability of constituents in any given single botanical product. Individual constituents in a botanical dietary supplement may impact drug action in both the pharmacokinetic and pharmacodynamic domains [9,14,20]. Factors including the part of the plant used, geographic location, time of harvest, age of plant, specific cultivar or species used, and secondary stressors may alter the nature of the secondary metabolites responsible for alteration of drug action [5,14]. The handling of crude plant product post-harvest, misidentification of species, and method of extraction may similarly impart significant inconsistency in product [14]. The combination of botanic species in formulated products only amplifies this variability [14]. Variations in processing may further complicate matters for both botanic and non-botanic dietary supplements, creating a wide variability in organoleptic characteristics, altering native chemical constituents, and sometimes adding potentially harmful solvent residues. Fortunately, the application of metabolomics in the pharmaceutical industry is an emerging technology that may help standardize the nature and identification of product characteristics in the future [26,27,28].

Reproducibility is the vanguard of “evidence-based” medicine [29]. The aforementioned variabilities in botanic product composition present major limitations to the reproducibility of studies of potential herb–drug interactions. In any experimental approach, authentication and phytochemical characterization are imperative to assure that the product being tested is appropriately identified and unadulterated, preferentially through the independent verification of labeled phytochemical content claims [14].

The approach to experimental evaluation is itself laden with factors that complicate the interpretation of results [8,14,30]. In vitro studies can offer valuable mechanistic insight into interactions, particularly those at the level of pharmacokinetic interaction. However, results from in vitro assays often diverge from the clinical realities of herb–drug interaction. Depending upon the study design, influences on drug disposition related to the induction of a transporter or enzyme may not be identified. The water solubility of many phytochemicals is poor, resulting in limited intestinal absorption and/or extensive first-pass metabolism; thus, in vitro assays often examine supraphysiologic concentrations of constituents [14,27]. The use of solubilizing agents and solvents, ionic influences, and changes in pH may alter the effects observed on the target enzymes and transporters [14,27]. In vitro studies examining the impact on metabolic enzymes cannot simulate the myriad of factors that may impact pharmacokinetic drug disposition due to factors such as phytochemical metabolites, interaction with the microbiome, protein binding, uptake/efflux transporters, and other factors. Similarly, those studies looking at the impact on transporters cannot simulate the impact from metabolic enzymes [14,30,31].

The use of animal models can offer a more realistic insight into the true nature of phytochemical and pharmaceutical interaction, but several limitations to this approach exist. Most case-reports of herbal–drug supplement interaction with drugs occur when people self-treat with such supplements; animal model testing often uses the parenteral administration of phytochemicals, thereby confounding impacts from the gut microbiome, limitations in oral bioavailability, and the pre-systemic formation of metabolites. Inter-species differences in metabolic enzymes and transporters confer a major limitation when translating results to clinical applicability [14].

Clinical studies of herb–drug interaction obviate many of the aforementioned confounders, yet the information gleaned remains bounded by limitation. Major shortcomings from observational studies include a lack of product authentication, substantial risk of bias and confounding from many sources, and inability to prove causation. Clinical trials with herbal dietary supplements are difficult to blind, are often performed on relatively homogeneous groups of healthy patients with strict inclusion and exclusion criteria, and ultimately need to be coupled with rigorous product analysis and authentication. Many studies have found that real-world herbal dietary supplement users often have complicating comorbidities and take multiple prescription drugs [6,7,8,9,10,11]. Polypharmacy itself is an indicator of risk for drug–drug and herb–drug interaction [9,16,32]. Polymorphisms in genes coding for transport proteins and enzymes may not be present in a relatively small homogenous population [5]; thus, such influences from gain-of-function mutation on herb and drug disposition will not be detected. Often, in real-world reports of herb–drug interaction, the overuse of the herbal supplement leads to an effect that might not occur with normal exposure levels [17]. Pharmacokinetic pathways for most constituents of herbal products are not known, and the influence of prescription drugs on herbal product disposition is often overlooked [5]. Additionally, statistically significant differences in pharmacokinetic parameters may not lead to clinically significant impact [5]. Quite often, clinical studies of the same botanical produce conflicting results when performed by different researchers, which is perhaps due in part to differences in product formulations [5].

Very few single-botanic products actually are associated with clinically significant drug interaction, and such an interaction may provide benefit rather than harm [8,14]. The application of artificial intelligence and Natural Language Processing offer exciting potential for risk evaluation; however, these tools are not generally available at the clinic level and still require significant labor for data extraction [20]. Pragmatic trials are needed to discern real-world risks and benefits of herbal supplement–drug interaction [8]. In addition to providing widely applicable and relevant outcome data, pragmatic trials are ideal to evaluate complex interventions applied in a multidisciplinary fashion [33]. Until such trials are available to inform clinical decision making, health-care providers must triangulate available information to discern the best approach to optimize patient-centered care for the burgeoning number of patients who elect to include herbal and other dietary supplements. Herb–drug interaction databases often derive their data from various types of studies with their accompanying limitations. Thus, providers must combine personal clinical experience, data from case reports, data from primary literature, and information from herb–drug interaction databases to evaluate potential benefits and risks.

This pilot study is a convenience sample survey-based study that was conceived as a descriptive analysis of real-life experiences with herb–drug interaction from the perspective of herbalists, licensed health-care providers, and lay persons. A disproportionality analysis was planned on herb–drug interacting pairs gleaned from respondents using reports from the last five years found in the U.S. Federal Adverse Event Reporting System (FAERS). Disproportionality analysis for the occurrence of the reaction specified for each reported herb in the herb–drug interaction was planned using data in the Center for Food Safety and Applied Nutrition (CFSAN) Adverse Event Reporting System (CAERS). The disproportionality analyses aimed to evaluate the ability of these national reporting systems to aid in parsing signal from noise during clinical evaluation of suspected dietary supplement–drug interaction adverse events, using tools available to most clinicians. Secondary aims of the study include the identification and evaluation of the most commonly used sources of herb–drug interaction information, exploration of the reasons for respondent use of dietary supplements, and qualitative analysis of respondent’s perceptions of dietary supplement–drug interaction.

## 2. Results

### 2.1. Summary Statistics from Survey

A total of 131 questionnaire responses were initiated; 108 questionnaires (82.4%) were filled to completion. Four of the incomplete questionnaires contained no data, and eighteen provided only information about status as health-care providers/herbalists (HCP/H). Ninety-three lay persons and thirty-four HCP/H initiated the survey. Figure 1 displays the credentials associated with the respondent’s role as they interacted with patients; several respondents were associated with more than one credential. Seven HCP/H provided the zip code where they observed most HDI, with three reporting from the mid-Atlantic states and one each from the southeastern US, western US, southwestern US, and New England. Too few respondents included zip code information, preventing meaningful subgroup analysis by geographic region.

Herbal dietary supplement (HDS) use was reported by 62.9% of all respondents, with 18 (66.7%) HCP/H reporting that they had ever used HDS and 50 (59.5%) lay persons reporting the same. Pearson’s chi-squared test determined that the difference in rates was not statistically significant (X^2^ = 0.4393, 110, *p* = 0.507474). Of those reporting HDS use, 13.2% reported having personally experienced and herb–drug interaction (HDI); 11.1% of HCP/H and 12.4% of lay persons reported such personal experience. More than one-fourth (28.6%) of HCP/H respondents reported having observed at least one HDI. While the survey intended to target adverse HDI, 33.3 % of all unique HDI reported (both experienced and observed) were reported as beneficial in nature. No respondent reported the use of a drug interaction scoring tool. Table 1 details the HDI experienced and observed. Parameters not specified by survey respondents are identified as “NS” and are presented in Table 1 to provide a complete quantification of survey results. For example, respondent 131 answered “yes” that they had experienced an NDI but did not disclose which supplement or drug and did not specify the reaction; this feature of solicited and spontaneous reports adds to the complexity of interpretation. To help quantify the frequency of observations, HCP were asked how often a particular reaction was observed, which is reflected by the italicized content in the fourth column when specified.

### 2.2. Reasons for Using HDS

Reasons for using HDS fell into fourteen thematic categories and one miscellaneous category. Decreased expense was the most commonly cited reason among laypersons, while autonomy/self-reliance was the most common theme among HCP/H. Many respondents offered multiple reasons; each reason was categorized into the appropriate thematic category. Table 2 depicts the distribution of response by thematic category and group.

### 2.3. Source of Knowledge of HDI Mechanisms

Twenty-four HCP/H responded to the question “Where did you learn about potential mechanisms of herb–drug interactions?”. Many respondents offered more than one source of information; each source was coded into an appropriate thematic category. The most frequent source of information was cited as school or formal training, with 58.3% acknowledging that they were informed of HDI mechanisms during training. Figure 2 depicts the thematic response as a percentage of all responses.

### 2.4. Herb–Drug Interaction Checkers and Disproportionality Analysis

Twenty-one HCP/H responded to the question, “Which resources or herb/supplement–drug interaction checking programs do you routinely use (if any) to evaluate suspected or potential interactions?” Lexicomp™ was the most frequently cited at six respondents reporting use. “Pharmacist” refers to asking a pharmacist. Figure 3 depicts the percentage of citations for all specific sources cited more than once. The category “other” includes four unspecified texts, two unspecified online interaction checkers, and one each for PubChem, Micromedex™, Examine.com, “ask an ND”, Herbal Contraindication and Drug Interactions by Brinker, inference from Medscape drug metabolism, inference from American Herbal Product Association Handbook, RxList, Little Herb Encyclopedia, and I Pro.

HDI reported by survey respondents that contained sufficient information (HDS, drug or specific drug class, type of reaction) were looked up by two authors (JVS, MBB) in the specific references cited by more than two respondents; “ask a pharmacist”, PubMed and Google were deemed too general for inclusion into the analysis. Figure 4 depicts the level of agreement between respondent reports and Drugs.com [34], Epocrates [35], Lexicomp™ [36], Natural Medicines Database (NMD) [37], and Stockley’s Herbal Medicine Interactions [38].

A disproportionality analysis was performed using the CAERS database [39] to evaluate the association of the reported adverse reaction with the HDS for respondent reports containing sufficient information to perform the evaluation. As adverse event reporting databases, beneficial interactions are not reported in the adverse even reporting system and thus were not evaluated. No case reports were found for *Rauwolfia serpentina* (snakeroot), *Medicago sativa* (alfalfa), or *Glycyrrhiza glabra* (licorice). As seen in Table 3, all evaluable herb–reaction pairs were associated with the reported reaction at the pre-defined level of significance.

A disproportionality analysis was performed using data from the last five years located in the FAERS database [40], which was accessed via the public dashboard. Six HDI reports by respondents presented sufficient information to perform the analysis. As the most commonly used HMG CoA reductase inhibitor [41], atorvastatin was selected from the HMG CoA reductase (“statin”) class to evaluate the HDI report involving red yeast rice. No reports involving escitalopram, St. John’s wort and serotonin syndrome or alfalfa, warfarin, and decreased INR or clotting were found. Three of the six evaluable HDI reports returned with a significant association, as seen in Table 4.

## 3. Discussion

This study found that 59.5% of all respondents used dietary supplements, which is a finding similar to the 2017–2018 National Health and Nutrition Examination Survey, which revealed that 57.6% of adults used dietary supplements [1]. Herb–drug interactions (HDI) were experienced by more than one in ten respondents. While the questionnaire intended to identify adverse HDI, 44.4% of HDI experienced and 26.3% of observed HDI were reported as beneficial. A rapid review of the literature published in PubMed in the last five years keyed to the MeSH term “herb drug interaction” found the theme of beneficial interaction reflected in 17 (4.8%) of the 354 articles identified.

Many publications covering herbal medicines dogmatically cite that people use herbs because of the misconception that “natural means safe” [21]. In this study, several respondents reported that they used HDS because they prefer a natural approach, and several cited the safety of HDS over prescription medications, but only one cited both: “natural vs synthetic seemed safer and healthier choice” (respondent 73, layperson). Reasons for HDS use were elicited by a question that asked, “What factors influenced you to use or recommend/prescribe herbal medicines?” Respondents preferred supplements because they perceived that supplements are superior to prescription drugs (28.3% of total responses). The reasons for this perception include themes related to natural/not synthetic nature of supplements (13.2% of total responses), increased safety or gentleness of supplements (9.4%), and the ability of supplements to produce effects not seen with prescription drugs (5.7%). As respondent 114 (RN, paramedic) stated, “botanical medicine can do things that single-entity drugs can’t”. Another overarching reason for HDS use reflects the status of health care in the United States (27.4%), with 17% reflecting decreased cost of HDS compared to prescription drugs and 7.5% reflecting mistrust of the medical establishment. Three laypersons specifically pointed at mistrust of physicians, with respondent 124 stating “doctors don’t want to fix what’s wrong because it decreases return visits”. While culture/family was cited as only 4.7% of total reasons, four out of five respondents who cited culture/family also cited mistrust of the medical establishment. The only cited reason with a statistically significant response rate difference between HCP/H and laypersons related to autonomy and choice; this was likely due to prescribers citing patient autonomy as a reason for including HDS in the regimen.

The questionnaire asked an open-ended question to elicit opinions and insights regarding herb–drug interactions. In response to this question, four informants responded with concerns about product quality: “It’s hard to know exactly what’s in a botanical that you buy off the shelf—lack of regulation means no certainty” (respondent 112, MD). Concerns about pesticide contamination were mentioned by three respondents: “It’s hard even in the country to get safe plants—they spray everywhere…” (respondent 122, lay person). As previously mentioned, to control for adulteration, the analysis and authentication of botanicals is a critical aspect of experimental study design. A rapid review of literature published in PubMed in the last five years keyed to the MeSH term “herb drug interaction” found 16 clinical studies investigating HDI. Of these, only 50% specified that product analysis and/or authentication was performed.

The most common theme that emerged from responses to this open-ended opinion question related to knowledge and communication between health-care providers and patients. Four respondents, all nurses, stressed the importance of consulting with a physician. Two respondents, both physicians, mentioned that providers often do not ask and patients often do not tell unless specifically asked about HDS use. Four HCP/H respondents reported concerns with a lack of provider knowledge of HDS or lack of research (two DO, one PA/DMS/DHSc, one RPh/PharmD). This sentiment is supported by a study of health-care providers in the United Kingdom, which found that 37% of participants were not aware of any interactions between HDS and prescription medications [32]. Regarding knowledge of Traditional Chinese Medicine (TCM), respondent 111 (DO) stated that “I have tried to learn TCM for years, it takes a vast amount of knowledge to function adequately”. Those possessing the “vast amount of knowledge needed to function adequately” are often not included during design and implementation of clinical trials, which further complicates the interpretation of the research that has been performed.

Evaluation of the common references used by HCP/H to evaluate HDI revealed several limitations in these tools. The search of one online database (Epocrates [35]) did not find any of the HDS reported by respondents. All other resources agreed with reports of bleeding from combinations of warfarin with cranberry and Ginkgo and combinations of garlic and aspirin. Disproportionality analysis for bleeding with the HDS alone revealed an association of each with this reaction. The disproportionality analysis from FAERS indicated a signal of disproportionality only with the combination of Ginkgo, warfarin, and bleeding. Similarly, all resources other than Epocrates [35] cited interaction between melatonin/eszopiclone causing hangover or excessive sedation and between St. John’s wort/escitalopram causing the serotonin syndrome; only the former was detected as a significant signal through disproportionality analysis using FAERS data. The Natural Medicine Online Database [37] was the only resource to agree with the report of Red Yeast Rice/statin causing muscle pain. Disproportionality analysis of FAERS data for this combination (using atorvastatin-related data as baseline) detected a significant signal of disproportionality for this interaction; CAERS also detected a signal of disproportionality for Red Yeast Rice alone in association with muscle pain.

Of the resources commonly used by respondents, perhaps not surprisingly, the two that focus primarily on herbal dietary supplements (Natural Medicine Database [37], Stockley’s Herbal Medicines Interactions [38]) demonstrated the greatest agreement with the reports of experienced and observed HDI. Of the three respondents relying on the Natural Medicine Database, [37] two were pharmacists, and one was a Naturopathic Physician (ND); the ND and one of the pharmacists were also the two to cite the latter resource.

In pharmacovigilance, signals of disproportionality do not establish causality, but rather signal a need for clinical evaluation. Four HDS–drug interacting pairs identified by respondents in this study were reflected with agreement from all HDI checking resources that report HDS interactions, with two of these pairs associated with a signal of disproportionality when evaluated against a large set of adverse event data. Thus, the use of FAERS to aid in the evaluation of clinically encountered HDI seems to lack sensitivity. Furthermore, handling the large dataset is likely too cumbersome for most in clinical practice, the multiplicity of items in each field of this database also necessitates additional manipulation to identify secondary search targets after initial search. The CAERS database does not report concomitant medications, so it cannot be used to evaluate HDS in combination with medications; the lack of this reporting may decrease the specificity of the disproportionality analysis. CAERS corresponds well with the reported HDI, allows for the filtering of more than one field simultaneously, does not limit the number of items in a search field, and can be evaluated with Excel™ alone. Performing a disproportionality analysis with this data could assist in the clinical evaluation of suspect HDI by detecting whether the reaction of interest is associated with a signal of disproportionality for a given HDS. The reporting of adverse events to each database is spontaneous; thus, neither can be used to calculate actual rates of event occurrence, nor does submission of a report constitute proof that the suspected product caused or contributed to the adverse event. A positive association of the suspected product and event also may indicate a tendency for their concomitant reporting related to confounding factors. As adverse event reporting tools, positive associations by definition are excluded. Furthermore, the data in these databases are devoid of clinical context; careful clinical evaluation is required to interpret the potential of an HDI [14].

As repositories of adverse events, neither database can detect positive signals of disproportionality. Two survey respondents (both laypersons) reported an additive beneficial interaction between *Cannabis* or its constituents and pain medications in improving relief of pain. The four commonly used HDI evaluation resources provided information discordant to these responses in that all reported a negative interaction with increased risk of CNS depression from this combination. Only one resource reflected positive interactions between licorice and gastric ulcer prevention from NSAIDs and the ability of coenzyme Q10 to reduce statin-related muscle pain. While additive effects are often reported as a negative, four survey respondents (one ND, one herbalist, one pharmacist, one layperson) reflected that this may actually be a beneficial interaction if managed appropriately. “I’m often leveraging alterations in drug metabolism caused by the herb/supplement to reduce the side effects or needed dose of the drug or increase efficacy of the given drug dose” (respondent 34, herbalist). This sentiment was echoed by the ND (respondent 2) who stated “…the interaction may make it so that the “side effect” is that they have to take less of the drug…another common drug/herb/supplement side effect that I see daily is that the herb/supplement diminishes or stops completely a problem off-target effect of a drug”. The collaboration between scientists, licensed health-care providers, and herbalists is needed to assess these potential benefits both through observational studies and pragmatic clinical trials. The potential contribution of repositories for adverse event reporting cannot be over-emphasized; we must all be diligent and responsible in reporting our clinical observations and experiences.

There are several major limitations to this study. As a small pilot study, the results from the questionnaire may not reflect the population as a whole. The small number of HCP/H respondents prevented meaningful subgroup analysis by credential and geographic region. As a descriptive study, the study was not designed to evaluate the specificity nor sensitivity of any HDI checking resources or database analyses. As seen in our survey results, reports of events often lack sufficient detail for meaningful analysis. The limitations of pharmacoepidemiologic approaches are numerous; the reader is referred to excellent reviews by Bate and Evans [42] and Faillie [43] for details regarding the limitations of this methodology.

## 4. Materials and Methods

### 4.1. Online Questionnaire

An online questionnaire was designed using the Qualtrics™ survey platform. The questionnaire included both open-ended and close-ended questions and was designed to be an anonymous, online, self-completion questionnaire launched on social media with snowball recruitment, wherein respondents were asked to re-post the invitation to their own social media pages. Age under 18 years was the only exclusion criteria for participation. No compensation was provided to respondents. The questionnaire and research plan were reviewed by the Lincoln Memorial University Institutional Review Board (IRB) and deemed exempt from IRB oversight. Raw data from the questionnaire will be provided upon request.

The first portion of the questionnaire targeted licensed health-care providers and those who recommend herbal dietary supplements for use in others (hereafter referred to as “herbalists”). Close-ended questions sought information on specific licensure/practice and whether the respondent had ever observed a suspected herbal supplement–drug interaction. Open-ended questions sought information on specialty focus of practice (if any), zip code where the most interactions (if any) were noted, specifics on the herb, drug, reaction, probability rating tool (if applicable) and rating score (if applicable) for the interaction(s), estimation of frequency with which the interaction was observed or suspected, sources of information used to evaluate suspected interactions, and where the respondent learned about mechanisms of herb/drug interactions.

The second portion of the questionnaire sought information about personal herbal dietary supplement use and experience with supplement–drug interactions from both lay persons and licensed health-care providers/herbalists. Closed-ended questions sought to discern herbal supplement users from non-users and in the supplement user subgroup, those who experienced a suspected herb–drug interaction from those who had not. Open-ended questions sought information on factors that influenced respondents to use/recommend herbal supplements, information about the herb, drug, and reaction if an interaction was experienced, and insights or opinions about herb/drug interactions. Information from incomplete surveys was included in the analysis. While the questionnaire aimed specifically at herbal (botanical) dietary supplements, many participants included non-botanical supplements; these were included in the analysis and are subsumed hereafter by the terminology “herbal dietary supplements (HDS)”, “herb”, and “herbal product”.

### 4.2. Evaluation of Herb–Drug Interaction Checking Resources

All herb–drug interaction resources that were cited by more than one respondent were accessed to evaluate any interaction reported by respondents that included sufficient specific information about the herb or supplement, drug or drug class, and type of reaction.

### 4.3. Disproportionality Evaluation of Herbal Product and Reaction from CAERS

The CAERs database (https://www.fda.gov/food/compliance-enforcement-food/cfsan-adverse-event-reporting-system-caers) (accessed on 14 December 2022) was searched for all reports filed between January 2004 and June 2022 was downloaded as an Excel™ spreadsheet by clicking the “download CAERS excel” button. An Excel™ formula command was applied to the REPORT_ID field to count all unique cases. Herbal product of interest was selected by filtering the PRODUCT field using both common names and scientific name. The CASE_MEDRA_PREFERRED_TERMS (reaction) field was filtered by searching the Medical Data Dictionary for Regulatory Activities (MedDRA) terms and synonyms for the reported reaction. The number of unique reports of the reaction of interest combined with the herbal product of interest (cases with herb, A) was counted. The number of unique reports for the reaction of interest without the specific herb (cases without herb, B) was calculated by removing the filter for the HDS, using an Excel™ command to count the unique reports, and subtracting A (the number cases with HDS). Non-cases with HDS (C) were calculated by removing the filter for the reaction field, counting the unique reports using an Excel™ command, and subtracting the number of unique reports with HDS (A). Non-cases without HDS (D) were calculated by subtracting the number of cases with HDS (A), the number of cases without HDS (B), and the non-cases with HDS (C) from the total number of unique reports in the database. Reporting Odds Ratio (ROR) was calculated using the formula (A × D)/(B × C), the standard error (SE) was calculated by taking the square root of the sum of the reciprocals of A, B, C, and D, and the 95% confidence interval was calculated by multiplying ROR by Euler’s number to the power of +/−1.96 × (SE) [40]. Significance for the signal of disproportionality was predefined as a lower limit of the 95% confidence interval > 1 and at least one case reported in association with the herb of interest. All calculations were performed in Excel™. Figure 5 offers a graphic overview of the calculation.

### 4.4. Disproportionality Analysis of Herbal Product with Drug with Reaction of Interest from FAERS

Data from FAERS database were retrieved between 18 and 23 December 2022 from the public dashboard (https://fis.fda.gov/sense/app/95239e26-e0be-42d9-a960-9a5f7f1c25ee/sheet/7a47a261-d58b-4203-a8aa-6d3021737452/state/analysis, accessed on 14 December 2022). The search function was used to identify all reports involving the victim drug in the past five years, using both trade and generic name. Combination products containing the victim drug with other prescription ingredients were not included. Where the number of generic names and trade names exceeded five (the maximum number of allowable items in FAERS), multiple searches were performed. Details of individual reports were downloaded into an Excel™ file. The results of multiple searches for a victim drug were compiled into a single file. Duplicate reports are removed in FAERS; thus, the identification of duplicates was not needed. The lexical analysis function of MAXQDA20 (Verbi software2020, v2020.1) was used to code each report for the presence or absence of the target herbal product and MedDRA-defined reaction of interest. Mixed method quantizing was used to count the presence or absence of the herb and reaction of interest; quantized results were exported into Excel™ files. Excel™ commands were used to count cases where the herb and reaction were present with the victim drug (A), where the herb and drug were present without the reaction of interest (B), where the reaction and drug were present without the HDS (C), and other adverse effect reports of the drug without the presence of the HDS (D). The reporting odds ratio, standard error, and 95% confidence intervals were calculated in Excel™ and predefined significance for signal of disproportionality as with the CAERS data.

### 4.5. Analysis of Survey Results

MaxQDA20™ software was used to code results from the survey. Variables included role as health-care provider/herbalist or layperson, credential or license under which the respondent practiced, and region where herb–drug interactions were observed as identified by respondent zip code using the USPS downloadable zip code file (https://www.downloadexcelfiles.com/us_en/download-zip-code-list-united-states-postal-service-usps#.Y6t9QRXMLrc) (accessed 14 December 2022). One author (MBB) reviewed open-ended texts to identify themes. Coded and sorted data were exported into Excel™ for analysis. Pearson’s chi square for comparison of reported HDS use between health-care provider/herbalist and lay persons was performed in Excel™. Fisher’s exact test was performed using the free internet calculator “Easy Fisher Exact Test Calculator” [44] to evaluate differences in frequencies of reported reasons for using HDS between HCP/H and lay persons. In each case, significance level was set at alpha ≤ 0.05.

## 5. Conclusions

This pilot study demonstrates that data from the FAERS database are too unwieldy to offer insight into the association of an herbal dietary supplement with a drug and adverse reaction. The CAERS data are accessible in a user-friendly fashion and may help establish the association of an herbal supplement with a given reaction but do not contain information about concomitant medications. Notwithstanding these limitations, these repositories are critical for signal detection; we must be diligent in our efforts to report our observations and experiences. Larger studies based upon solicited reports are needed to further identify the prevalence of and experience with HDI. Resources commonly used by participants to evaluate herb–drug interactions often do not report HDI adequately, particularly those resources that focus primarily on prescription drugs. Participants in the study often cite potential beneficial herb–drug interactions that are reflected neither by herb–drug interaction resources nor adverse event reporting databases; more research is needed to evaluate these potential beneficial interactions. Collaboration and communication are vital in both researching causality of herb–drug interactions and evaluating the clinical benefits and risks during patient care. All members of the team need to be involved in the discourse, including the patient. As respondent 3 (DO) states, “Patients are still hesitant to disclose this information unless specifically asked; it is so important to know and understand our patients”.

## Figures and Tables

**Figure 1 medicines-10-00020-f001:**
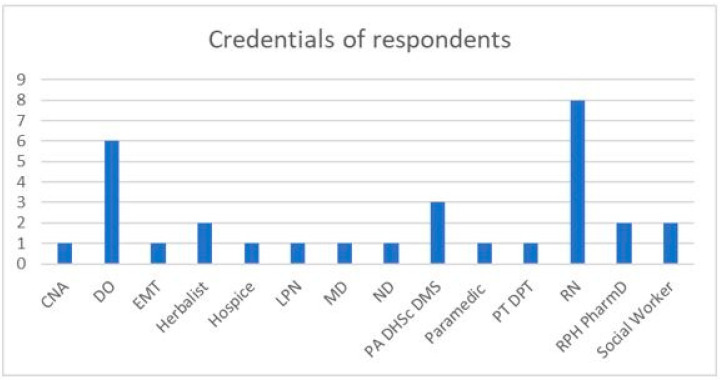
Credentials associated with respondent’s roles in patient interaction. CNA Certified Nursing Assistant; DO Doctor of Osteopathy; EMT Emergency Medical Technician; LPN Licensed Practical Nurse; MD Allopathic Medical Doctor; ND Naturopathic Doctor; PA/DHSc/DMS Physician Assistant; RPh/PharmD pharmacist.

**Figure 2 medicines-10-00020-f002:**
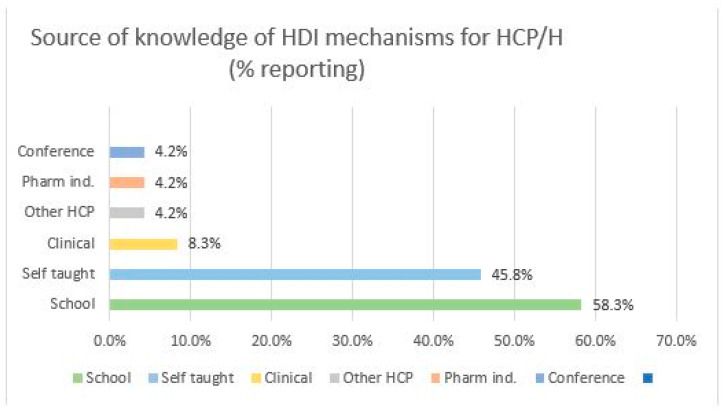
Source of knowledge among HCP/H. Pharm ind.: Pharmaceutical industry or groups; Other HCP: Other health-care provider; Clinical: Clinical experience.

**Figure 3 medicines-10-00020-f003:**
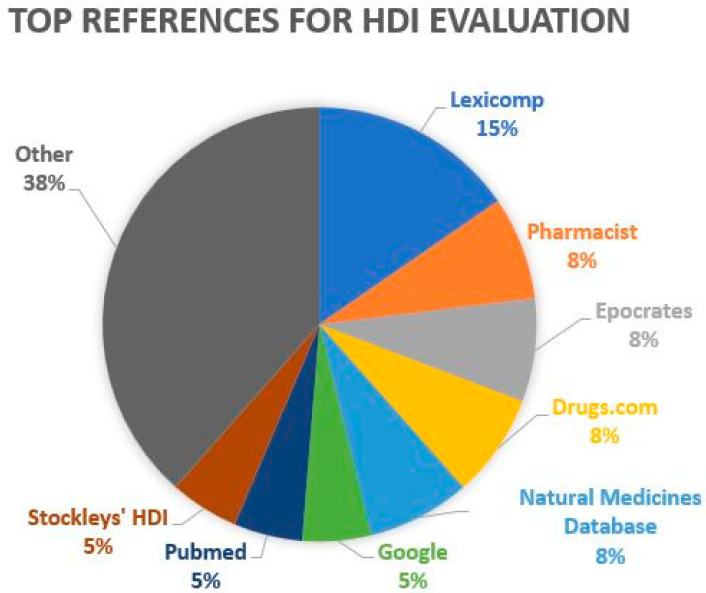
Common references used to evaluate HDI.

**Figure 4 medicines-10-00020-f004:**
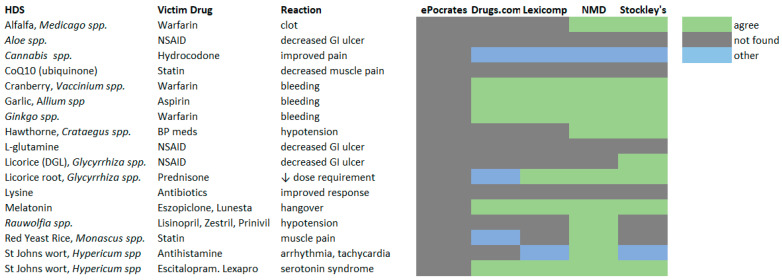
Agreement between HDI checking resources and respondent reports. Agree—HDI resource-matched herb–drug or drug class pair and reaction; not found—no herb–drug reaction found for the pair; other—an unrelated reaction found for the pair. BP blood pressure, NSAID non-steroidal anti-inflammatory drug NMD Natural Medicines Database.

**Figure 5 medicines-10-00020-f005:**
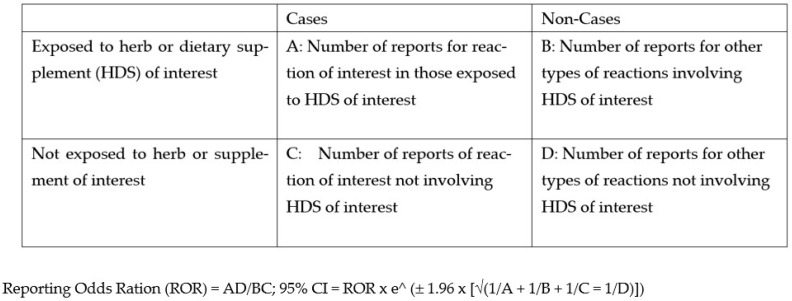
Method of calculating reporting odds ratio (ROR) and confidence intervals.

**Table 1 medicines-10-00020-t001:** Herb/Supplement Drug Interactions (HDI) reported in survey.

**Herb/Supplement Drug Interactions Experienced by Respondents**			
**HDS**	**Drug**	**Reaction**	**Credential (response ID)**
*Cannabis*	Hydrocodone	Improved pain relief *	Layperson (107)
*Cannabis*	Painkillers	Improved pain relief *	Layperson (104)
Coenzyme Q 10	Statin	Reduced muscle pain *	Layperson (118)
Lysine	Antibiotic	Improved response *	Layperson (118)
Melatonin	Lunesta^®^	Hang-over	MD (112)
NS	NS	NS	Layperson (131)
NS	NS	NS	ND (2)
Red yeast rice	Statin	Muscle pain	Layperson (113)
St John’s wort	Antihistamine	Tachycardia	Layperson (117)
**Herb/Supplement Drug Interactions Observed by Respondents**			
**HDS**	**Drug**	**Reaction**	**Credential (ID), *frequency***
Alfalfa	Warfarin	NS	DO (30)
Aloe	NSAID	Reduced gastric ulcer *	ND (2)
Cranberry	Warfarin	Bleeding	RPh/PharmD (77), × 2
Crataegus	BP meds	Hypotension	RPh/PharmD (77), *several*
Fish oil	Psych meds	NS	DO (3)
Garlic	Aspirin	NS	DO (3)
Ginger	Chemotherapy	Reduced nausea *	ND (2)
Ginkgo	Warfarin	Bleeding	RPh/PharmD (77), *many*
Leeks	Warfarin	Decreased INR	RPh/PharmD (77), × 5
L-glutamine	NSAID	Reduced gastric ulcer *	ND (2)
Licorice (DGL)	NSAID	Reduced gastric ulcer *	ND (2)
Licorice root	Prednisone	Reduced dosing need *	Herbalist (34)
NS	NS	Elevated INR	RN (78)
*Rauwolfia*	Lisinopril	Additive effects	ND (2)
Salt substitute	ACE inhibitors	Hyperkalemia	RPh/PharmD (77) × 2
Saw palmetto	Antibiotics	NS	DO (25)
St John’s wort	NS	NS	DO (25)
St John’s wort	Escitalopram	Serotonin syndrome	RPh/PharmD (77)
Thyroid support formula	Levothyroxine	NS	DO (30)

* Beneficial interaction; NS not specified, DO Osteopathic Physician, ND Naturopathic Physician, RPh/PharmD Pharmacist, RN Registered Nurse. Frequency corresponds to HCP respondent report of how often the particular interaction was observed. “Thyroid support formula” is a non-specific product reported by one respondent.

**Table 2 medicines-10-00020-t002:** Reasons for HDS use. Percent total represents the number of responses in thematic categories compared to total number of responses.

Thematic Reason	LP	HCP/H	Total	(%)	*p*
Less Expensive	14	4	18	16.8	0.19
Natural, less exposure to synthetics	10	4	14	13.1	0.57
Safer, gentler, fewer adverse effects	5	5	10	9.3	0.49
Evidence, experience, logical choice	4	4	8	7.5	0.46
Lack of trust in medical establishment	6	2	8	7.5	0.71
Benefit, to treat specific condition	4	3	7	6.5	0.71
Self-reliance, choice, autonomy	1	6	7	6.5	0.01 *
Global effects, effects not seen from medications	2	4	6	5.6	0.19
Culture, family	4	1	5	4.7	0.65
Recommended by other	5	0	5	4.7	0.16
Accessibility	3	1	4	3.7	1
Efficacy	4	0	4	3.7	0.29
Avoid/reduce number of prescription medications	2	2	4	3.7	0.62
Curiosity, open-mindedness	1	2	3	2.8	0.55
Other	2	1	3	2.8	1

* Significant at alpha 0.05; HCP/H health care provider/herbalist; LP layperson.

**Table 3 medicines-10-00020-t003:** Disproportionality analysis for HDS with reaction of interest. A: Number of reactions of interest in those exposed to herb of interest; B: Number of other types of reactions reported for herb of interest; C: Number of reactions of interest reported without herb of interest; D: Number of other types of reactions reported without herb of interest. ROR reporting odds ratio = AD/BC; CI confidence interval, * statistically significant at alpha 0.05.

HDS	Reaction	A	B	C	D	ROR	95% CI
*Vaccinium macrocarpon* (Cranberry)	Bleeding	34	243	164	69,675	59.4	40.2–87.8 *
*Crataegus spp*. (Hawthorne)	Hypotension	2	15	400	69,699	23.2	5.3–101.9 *
*Allium sativa* (Garlic)	Bleeding	40	259	193	69,624	55.7	38.8–80.0 *
*Ginkgo biloba* (Ginkgo)	Bleeding	30	115	179	69,792	101.7	66.3–156.0 *
Melatonin	Hangover	13	79	260	69,764	44.1	24.2–80.4 *
*Monascus purpureus* (Red Yeast Rice)	Muscle pain	35	120	424	69,537	47.8	32.4–70.5 *
*Hypericum spp*. (St John’s wort)	Tachycardia	5	49	2387	67,675	2.9	1.2–7.3 *
*Hypericum spp*. (St John’s wort)	Serotonin syndrome	5	49	1465	68,597	4.8	1.9–12.0 *

**Table 4 medicines-10-00020-t004:** Disproportionality analysis of FAERS data. A: Number of reactions of interest in those exposed to herb of interest; B: Number of other types of reactions reported for herb of interest; C: Number of reactions of interest reported without herb of interest; D: Number of other types of reactions reported without herb of interest. ROR reporting odds ratio = AD/BC; CI confidence interval; * statistically significant at alpha = 0.05.

HDS and Drug	Reaction	A	B	C	D	ROR	95% CI
*Allium sativum* (garlic) and aspirin	Bleeding	4	1	21,712	26,474	4.9	0.6–43.6
*Ginkgo biloba* and warfarin	Bleeding	212	8	559	23,872	1131.7	555.9–2303.9 *
Melatonin and eszopiclone	Hangover	18	81	676	26,235	8.6	5.1–14.4 *
*M. purpureus* (Red yeast rice) and atorvastatin	Muscle pain	3	3	5821	30,641	5.3	1.1–26.1 *
*V. macrocarpon* and warfarin	Bleeding	3	4	7885	12,684	1.2	0.27–5.4

## Data Availability

Data from FAERS and CAERS are freely available. Raw data from surveys are available upon request.

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
