# Peer review of "Everyday Evaluation of Herb/Dietary Supplement–Drug Interaction: A Pilot Study"

_medicines, 2023, doi:10.3390/medicines10030020_

Round 1
Reviewer 1 Report
Thank you for giving me the opportunity to read your article again. It is a subject that interests me on certain points. I will try to make some remarks to improve your document.
I find the introduction interesting but I think I have already read it several times. On the other hand, at the beginning of page 3 you do not mention that the industrial manufacturing process modifies the chemical composition of products. The same applies to the type of extraction used to make the food supplement. This is very important. I invite you to read the literature in metabolomics of herbal products that shows this and to mention them in the text.
At the end of your introduction, I think it would be good for you to mention that artificial intelligence and NLPs are valuable tools that will help in the closed future in the "quick" reading of the literature to identify HDIs. Again, I invite you to mention this with references.
I find your title inappropriate. It does not really reflect your study which is not only interested in herbs but also in other "natural" products (co-enzyme, lysine, melatonin, fish oil, thyroid support formula? -table 1). If you want to keep the title, remove those products that are not from plants.
I am not familiar with this last product (thyroid support formula) which to me is meaningless and should be explained. In the USA this product may be well known but not in Europe. Furthermore, your tables and figures are not of very good quality and should be worked on a little more. This is a pity because the text seems to me to be good.
Indeed, in table 1, is it useful to mention the ID of the response since we do not have access to the raw data. What is the point of making rows with NS mentioned (without saying what NS corresponds to). What does DO/ND/RPh mean? Why do you indicate "many" on the line ginkgo and warfarin when you only indicate one ID=77? WHO recommends to talk about drugs by indicating the NINs and not the trade names. Lexapro(r) should not appear in my opinion. The font in tables is too big. The readability of table 2 is poor in the version I downloaded. Same for figure 2 etc.... The legends need to be completed much more carefully. Why underline in the text the Stockley and not the other databases.
You do not say what corresponds to A, B, C, D ROR... except in the material and method section. This should be corrected and appear in the legends. The web page mentioned in section 4.3 is not accessible. It is difficult in this case to check your data. Figure 1 appears after the others with a quality to be reviewed++.
In the discussion, you mention that the databases are incomplete. You say that they do not provide information on beneficial activities. But this is just not the purpose of these databases to my knowledge.
You can say that they are not up to date, but not that they are missing data that is not in the score, in my opinion.
I also think you should discuss the under-reporting to autorithies of adverse reactions to HDIs...
Finally, for me the format of the references needs to be reviewed for refs 2, 3, 8, 16, 23, 25, 26, 28
Thank you for considering my comments.
Author Response
Thank you so kindly for the constructive recommendations; they add greatly to the quality of this manuscript. We are pleased that the concepts in the introduction are very familiar to one with experience in this topic and hope that it serves appropriately to inform those who lack this depth of familiarity.
We have changed the title to include dietary supplements, thank you for helping us to be more concise in this regard: "Everyday Evaluation of Herb/Dietary Supplement-Drug Interaction: a Pilot Study." We updated the language throughout the document to be consistent with the nomenclature herb/dietary supplement.
Per your comment, we added a referenced sentence on page 3 to include that the manufacturing process can alter both botanical and food supplements, including actual chemical composition:
"Variations in processing may further complicate matters for both botanic and non-botanic dietary supplements, creating wide variability in organoleptic characteristics, altering native chemical constituents, and sometimes adding potentially harmful solvent residues. Fortunately, the application of metabolomics in the pharmaceutical industry is an emerging technology that may help standardize the nature and identification of product characteristics in the future [27-29]".
The nature of solicited/spontaneous reporting unfortunately complicates interpretation. In our attempt at workaday analysis, our study involved components of such reporting, for clarity in quantification we included items that were either non-specified (NS) or reported vaguely (e.g., "thyroid support formula"). We appreciate that clarifying these points helps offer context to readers who may someday report a reaction and further exemplifies the difficulties with clinical interpretation of such reports. We clarified our reasoning in the text and in the legends for the associated tables. An example of clarification is found beginning at line 224.
Similarly, application of NLP and AI are exciting additions to tools that aid in parsing true signal from noise regarding such reports. We added a reference from Zhang et al. (2022) explaining both the potential benefit and the difficulties in the context of clinical practice: "Application of artificial intelligence and Natural Language Processing offer exciting potential for risk evaluation; however, these tools are not generally available at the clinic level and still require significant labor for data extraction [20]"
We offer access to raw data for those with sufficient interest to request access, thus opted to include respondent ID should any have such interest.
We also thank you for the recommendation to improve the quality of graphics. Per the attached, all graphics have been improved per your recommendations; figure 5 was renumbered as it was erroneously labeled in the draft. We updated all graphics with legends to explain abbreviations, and also included an explanation for the quantifiers for HCP/H experience that were previously included without explanation.
The link to the website in 4.3 requires the user to download an excel table; we clarified this in the text to improve the reproducibility of our data.
We reiterated that the adverse events reporting databases report adverse event unilaterally, therefore are not useful to explore beneficial reactions. We added two sentences to strengthen emphasis that reporting is essential (beginning at lines 446 and 566). We further reviewed formatting of all references, and corrected the format of the text reference Stockley's Guide to Herbal Interactions throughout the manuscript.
Again, your excellent suggestions provided much-welcome guidance to improve the quality of this manuscript. Please accept our heartfelt gratitude for sharing your expertise and time.

Reviewer 2 Report
A pilot survey-based study on herb-drug interactions in real-life descriptive design and setting, analyzed and reported against publicly available data bases (in USA). As a pilot study, this is a valuable contribution in showing problems and limitations in identifying real-life HDIs, or demonstrating poor performance of current data bases. Perhaps more clearly outlined suggestions as how to improve or correct the current inadequate situation would have been valuable.
Author Response
Thank you so kindly for your time and expertise in reviewing the manuscript originally titled "Everyday Evaluation of Herb-Drug Interactions: a Pilot Study". Per the attached manuscript, we offer a few more suggestions that might help mitigate the problems. Please accept our heartfelt gratitude for your insights that aided greatly in improving this document.
